# Uncertainty-aware deep learning in healthcare: A scoping review

Tyler J. Loftus[1,2]*, Benjamin Shickel[3], Matthew M. Ruppert[2,4], Jeremy A. Balch[1], Tezcan Ozrazgat-Baslanti[2,4], Patrick J. Tighe[5], Philip A. Efron[1,2], William R. Hogan[6], Parisa Rashidi[2,7], Gilbert R. Upchurch, Jr.[1], Azra Bihorac[2,4]

1 Department of Surgery, University of Florida Health, Gainesville, Florida, United States of America,
2 Intelligent Critical Care Center, University of Florida, Gainesville, Florida, United States of America,
3 Department of Biomedical Engineering, University of Florida, Gainesville, Florida, United States of America,
4 Department of Medicine, University of Florida Health, Gainesville, Florida, United States of America,
5 Departments of Anesthesiology, Orthopedics, and Information Systems/Operations Management, University of Florida Health, Gainesville, Florida, United States of America, 6 Department of Health Outcomes & Biomedical Informatics, College of Medicine, University of Florida, Gainesville, Florida, United States of America, 7 Departments of Biomedical Engineering, Computer and Information Science and Engineering, and Electrical and Computer Engineering, University of Florida, Gainesville, Florida, United States of America

* tyler.loftus@surgery.ufl.edu

**Data Availability Statement:** All data are in the manuscript and/or supporting information files.

**Funding:** T.J.L. was supported by the National Institute of General Medical Sciences of the National Institutes of Health under Award Number K23 GM140268 and by the Thomas Maren Junior

## Abstract

Mistrust is a major barrier to implementing deep learning in healthcare settings. Entrustment could be earned by conveying model certainty, or the probability that a given model output is accurate, but the use of uncertainty estimation for deep learning entrustment is largely unexplored, and there is no consensus regarding optimal methods for quantifying uncertainty. Our purpose is to critically evaluate methods for quantifying uncertainty in deep learning for healthcare applications and propose a conceptual framework for specifying certainty of deep learning predictions. We searched Embase, MEDLINE, and PubMed databases for articles relevant to study objectives, complying with PRISMA guidelines, rated study quality using validated tools, and extracted data according to modified CHARMS criteria. Among 30 included studies, 24 described medical imaging applications. All imaging model architectures used convolutional neural networks or a variation thereof. The predominant method for quantifying uncertainty was Monte Carlo dropout, producing predictions from multiple networks for which different neurons have dropped out and measuring variance across the distribution of resulting predictions. Conformal prediction offered similar strong performance in estimating uncertainty, along with ease of interpretation and application not only to deep learning but also to other machine learning approaches. Among the six articles describing non-imaging applications, model architectures and uncertainty estimation methods were heterogeneous, but predictive performance was generally strong, and uncertainty estimation was effective in comparing modeling methods. Overall, the use of model learning curves to quantify epistemic uncertainty (attributable to model parameters) was sparse. Heterogeneity in reporting methods precluded the performance of a meta-analysis. Uncertainty estimation methods have the potential to identify rare but important misclassifications made by deep learning models and compare modeling methods, which could build patient and

Investigator Fund. T.O.B. was supported by the National Institute of Diabetes and Digestive and Kidney Diseases of the National Institutes of Health grant K01 DK120784, R01GM110240 from the National Institute of General Medical Sciences, and by UF Research AWD09459 and the Gatorade Trust, University of Florida. P.T.J. was supported by R01GM114290 from the NIGMS and R01AG121647 from the National Institute on Aging (NIA). PR was supported by National Science Foundation CAREER award 1750192, P30AG028740 and R01AG05533 from the NIA, 1R21EB027344 from the National Institute of Biomedical Imaging and Bioengineering (NIBIB), and R01GM-110240 from the NIGMS. A.B. was supported by W. Martin Smith Interdisciplinary Patient Quality and Safety Award (IPQSA), Sepsis and Critical Illness Research Center Award P50 GM-111152 from the National Institute of General Medical Sciences, R01 GM110240 from the National Institute of General Medical Sciences, and by UF Research AWD09458. The content is solely the responsibility of the authors and does not necessarily represent the official views of the National Institutes of Health.

**Competing interests:** The authors declare no conflicts of interest.

clinician trust in deep learning applications in healthcare. Efficient maturation of this field will require standardized guidelines for reporting performance and uncertainty metrics.

## Author summary

Deep learning prediction models perform better than traditional prediction models for several healthcare applications. For deep learning to achieve it's greatest impact on healthcare delivery, patients and providers must trust deep learning models and their outputs. This article describes the potential for deep learning to earn trust by conveying model certainty–the probability that a given model output is accurate. If a model could convey not only it's prediction but also it's level of certainty that the prediction is correct, patients and providers could make an informed decision to incorporate or ignore the prediction. The use of uncertainty estimation for deep learning entrustment is largely unexplored, and there is no consensus regarding optimal methods for quantifying uncertainty. Our purpose is to critically evaluate methods for quantifying uncertainty in deep learning for healthcare applications and propose a conceptual framework for specifying certainty of deep learning predictions. We systematically reviewed published scientific literature and summarized results from 30 studies, and found that uncertainty estimation methods have the potential to identify rare but important misclassifications made by deep learning models and compare modeling methods, which could build patient and clinician trust in deep learning applications in healthcare.

## Introduction

Deep learning is increasingly important in healthcare. Deep learning prediction models that leverage electronic health record data have outperformed other statistical and regression-based methods [1,2]. Computer vision models have matched or outperformed physicians for several common and essential clinical tasks, albeit in select circumstances [3,4]. These results suggest a potential role for clinical implementation of deep learning applications in health care.

Mistrust is a major barrier to clinical implementation of deep learning predictions [5,6]. Efforts to restore and build trust in machine learning have focused primarily on improving model explainability and interpretability. These techniques build clinicians' trust, especially when model outputs and important features correlate with logic, scientific evidence, and domain knowledge [7,8]. Another critically important step in building trust in deep learning is to convey model uncertainty, or the probability that a given model output is inaccurate [8]. Deep learning models that typically perform well make rare but egregious errors [9]. If a model could calculate the uncertainty in its predictions on a case-by-case basis, patients and clinicians would be afforded opportunities to make safe, effective, data-driven decisions regarding the utility of model outputs, and either ignore predictions with high uncertainty or triage them for detailed, human review. Unfortunately, there is a paucity of literature describing effective mechanisms for calculating model uncertainty for healthcare applications, and no consensus regarding best methods exists.

Our purpose is to critically evaluate methods for quantifying uncertainty in deep learning for healthcare applications and propose a conceptual framework for optimizing certainty in deep learning predictions. Herein, we perform a scoping review of salient literature, critically

evaluate methods for quantifying uncertainty in deep learning, and use insights gained from the review process to develop a conceptual framework.

## Materials and methods

Article inclusion is illustrated in **Fig 1**, a PRISMA flow diagram. We searched Embase, MEDLINE, and PubMed databases, chosen for their specificity to the healthcare domain, for articles with "deep learning" and "confidence" or "uncertainty" in the title or abstract and for articles with "deep learning" and "conformal prediction" in the title or abstract, identifying 37 unique articles. Two investigators independently screened all article abstracts for relevance to review objectives, removing three articles. Full texts of the remaining 34 articles were reviewed. Study quality was independently rated by two investigators using quality assessment tools specific to the design of the study in question (available at: https://www.nhlbi.nih.gov/health-topics/study-quality-assessment-tools). Only studies describing healthcare applications that were good or fair quality were included in the final analysis, which removed four articles, leaving 30 total articles in the final analysis. Data extraction was performed according to a modification of CHARMS criteria, which included methods for measuring uncertainty in deep learning predictions [10]. The search was performed according to Preferred Reporting Items for Systematic Reviews and Meta-Analyses extension for Scoping Reviews (PRISMA-ScR) guidelines, as listed in **S1 PRISMA** Checklist.

During screening, there were disagreements between the two investigators regarding the exclusion of five articles; all disagreements were resolved by discussion of review objectives without a third-party arbiter. Cohen's kappa statistic summarizing interrater agreement regarding article screening was 0.358 (observed agreement = 0.848, expected agreement = 0.764), suggesting that screening agreement between reviewers was fair [11,12]. During full text review, there was a disagreement between the two investigators regarding the exclusion of one article, which was resolved by discussion of review objectives without a third-party arbiter. Cohen's kappa statistic summarizing interrater agreement regarding full text review could not be calculated because both observed and expected agreement were 0.964, but this high value suggests that agreement between reviewers was substantial.

## Results

Included articles are summarized in **Table 1**. Notably, the use of uncertainty estimation in these articles was rarely applied to building trust in deep learning among patients, caregivers, and clinicians. Therefore, the presentation of results will focus primarily on the content of the articles, and opportunities to use uncertainty-aware deep learning to build trust will be discussed further in the Discussion section as a novel application of established techniques.

Among 30 included studies, 24 described medical imaging applications and six described non-imaging applications; these categories are evaluated and reported separately. First, important themes from included articles are synthesized into a conceptual framework.

### Conceptual framework for optimizing certainty in deep learning predictions

Deep learning uncertainty can be classified as epistemic, (i.e., attributable to uncertainty regarding model parameters or lack of knowledge), or aleatoric (i.e., attributable to stochastic variability and noise in data). Epistemic and aleatoric uncertainty have overlapping etiologies, as variability and noise in data can contribute to uncertainty regarding optimal model parameters and knowledge regarding ground truth. In addition, epistemic and aleatoric uncertainty may be amenable to similar mitigation strategies, as collecting and analyzing more data may

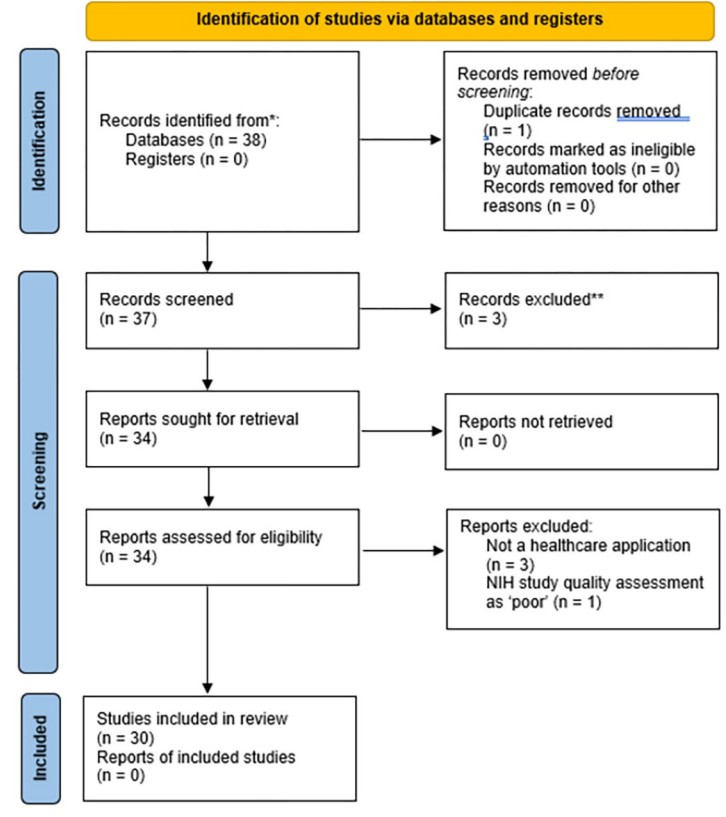

**Fig 1. PRISMA flow diagram for article inclusion.**

allow for more effective identification and imputation of outlier and missing values, reducing aleatoric uncertainty, and may also allow for more effective parameter searches. Beyond these overlapping etiologies and mitigation strategies, epistemic and aleatoric uncertainty have some unique and potentially important attributes. Epistemic uncertainty can be seen as a lack of information about the best model and can be reduced by adding more training data [13]. Learning curves stratified by number of training samples offer an intuitive approach to visualizing epistemic uncertainty, where it becomes evident that using more data typically results not only in more accurate models, but also in more stable loss when trained for the same number of epochs. In stochastic models, parameter estimates also become more stable with increasing amounts of training data. In addition to increasing knowledge through larger sample sizes, it may also be possible to reduce epistemic uncertainty by adding input features, especially multi-modal features (e.g., using not only vital signs to predict hospital mortality, but also using laboratory values, imaging data, and unstructured text data from notes written by clinicians), or modifying the algorithm to learn from additional nonlinear combinations of

**Table 1. Summary of included studies, classified as imaging or non-imaging applications.**

| Primary author | Purpose | Population or sampling unit | Sample size | Model architecture | Best model performance | Validation method | Method for quantifying prediction uncertainty | Quality Rating |
|---|---|---|---|---|---|---|---|---|
| *Medical imaging applications* | | | | | | | | |
| Araújo (34) | Grade diabetic retinopathy severity | Datasets of retinal images | Approximately 93,000 images | Convolutional-batch normalization blocks, max-pooling layers | Quadratic-weighted Cohen's kappa 0.71–0.84 for predictions vs. ground truth | External | Calculate Cohen's kappa statistics for model predictions at threshold levels of uncertainty, calculated by variance in image-wise retinopathy grade probability | Good |
| Athanasiadis (20) | Correlate visual and audio emotional expression | Audio-visual emotion datasets | 187 people, 7356 audio recordings, 7442 videos, 96 images | Generative Adversarial Networks | Classification 52.52% in one dataset and 47.11% in the other | External | Conformal prediction to obtain error calibration | Good |
| Ayhan (31) | Diagnosing diabetic retinopathy | Fundus images | 89,215 images | Convolutional neural network | AUC 0.959–0.982 | External | Calculate variance in the form of entropy as a distribution of predicted probabilities | Good |
| Cao (32) | Classify breast masses, identify tumors | Breast ultrasound images | 107 patients with 13,382 ultrasound slices | Dense U-Net | Accuracy 99.21% | Internal | Generate visual epistemic uncertainty maps for each image | Fair |
| Carneiro (29) | Classifying colorectal polyps | Images of colorectal polyps obtained by colonoscopy | 940 images from 287 patients | Residual and densely connected convolutional networks | Accuracy 0.76 | External | Classification entropy or the predicted variance produced by Bayesian methods | Fair |
| Edupuganti (35) | Quantify uncertainty in deep MRI segmentation | Knee MRI images | 19 patients with 320 2D image slices per patient | Variational autoencoders, convolutional neural networks | $R^2 = 0.97$ for 2-fold under sampling | External | Generate a posterior of the MRI image and generate pixel variance maps using Monte-Carlo sampling | Good |
| Graham (21) | Label regions and sub-regions of the brain | Brain MRI images | 593 scans | 3D UNet | Dice score 0.845 for all regions in uncertainty-aware hierarchical model | External | Cross-entropy uncertainty measured at each progressively smaller sub-region of the brain | Good |
| Herzog (15) | Diagnose ischemic stroke | Brain MRI images | 511 patients with average 30 images per patient | Bayesian convolutional neural network | Accuracy 95.9%, was 2% better than model without uncertainty measurements | Internal | Variance, variation ratio, and predictive entropy of a distribution of Bayesian probabilities | Good |
| Hu (30) | Diagnose a rare lymphoma | Positron emission tomography and computed tomography scan images | 83 patients | Convolutional neural networks, coarse-to-fine segmentation | Sensitivity 74.7% | Internal | Zone-based uncertainty estimates based on Monte Carlo dropout technique comparing the lesion and the background | Good |
| Ktena (22) | Evaluate similarity between functional brain networks | Brain functional MRI images | 871 subjects | Convolutional neural networks | Overall classification improvement with proposed metric 11.9% and AUC 0.58 | External | Calculate similarity between irregular graphs rather than calculating uncertainty directly | Good |

*(Continued)*

**Table 1.** (Continued)

| Primary author | Purpose | Population or sampling unit | Sample size | Model architecture | Best model performance | Validation method | Method for quantifying prediction uncertainty | Quality Rating |
|---|---|---|---|---|---|---|---|---|
| Lee (43) | Quantify uncertainty in brain metabolite identification | MRI, proton magnetic resonance spectroscopy | 15 rats | Convolutional neural networks | Measurement uncertainty for five major metabolites was less than 10% | Internal | Calculate Cramer-Rao-lower-bounds statistics to estimate the reliability of fitting | Fair |
| Leibig (44) | Diagnose diabetic retinopathy | Fundus images | 89,902 images | Convolutional neural networks | >85% sensitivity and 80% sensitivity when referring 20% of the most uncertain decisions for further inspection | External | Draw Monte Carlo samples from the approximate predictive posterior, use its standard deviation to represent uncertainty | Good |
| McKinley (45) | Detect multiple sclerosis lesion changes | MRI images | Training: 4–5 sets of 176 images for 26 patients, testing: 77 image sets | Convolutional neural networks | Accuracies of 75% and 85% in separating stable and progressive time-points | External | Use best-practice standards to annotate lesions, predict the probability that a convolutional neural network will assign a different label than assigned a ground truth | Good |
| Nair (36) | Detect multiple sclerosis lesions | MRIs from patients with relapsing-remitting multiple sclerosis | 1064 patients, annual MRIs during a 24-month period | Convolutional neural network | Overall lesion-level true positive rate of 0.8 at 0.2 false detection rate | External | Approximate probability distributions with Monte Carlo dropout and measure their variance, predictive entropy, and mutual information | Good |
| Natekar (37) | Classify brain tumors | Brain MRI images | Training: 285 cases, testing: 48 volumes | Convolutional neural networks | Whole tumor Dice coefficient 0.830 | External | The mean of the variance in a predicted posterior distribution generated by running a model for 100 epochs for each image | Fair |
| Qin (16) | Estimate brain and cerebrospinal fluid intracellular volume | Brain diffusion MRI scans | Approximately 1,000,000 images (not specified fully) | Convolutional neural network | All correlations between estimation uncertainty and error were significant ($p<0.001$) | External | Train an ensemble of deep networks, measure variance in their fused results | Good |
| Roy (46) | Identify brain structures | Brain MRIs | Four datasets with MRIs from 30, 29, 13, and 18 subjects | Convolutional neural network | Dice = 0.88, 0.83, 0.81, 0.81 | External | Samples are passed through the neural network serially, some weights dropped each time, derive voxel-wise and structure-wise uncertainty from variance across runs | Good |
| Sedghi (23) | Model agreement for brain image classifications | Brain MRIs | 115 subjects | Convolutional neural network | Intra-subject dice for gray matter, white matter, cereprospinal fluid = 0.70, 0.77, 0.62 | External | Calculate variance in displacements for different image classifications | Good |

*(Continued)*

**Table 1.** (Continued)

| Primary author | Purpose | Population or sampling unit | Sample size | Model architecture | Best model performance | Validation method | Method for quantifying prediction uncertainty | Quality Rating |
|---|---|---|---|---|---|---|---|---|
| Seebock (38) | Detect anomalies in retinal optical coherence tomography images | Optical coherence tomography B-scans | 226, 33, 31 | Bayesian U-Net, convolutional neural network-based | Precision = 0.748, recall = 0.844, Dice = 0.789 | External | Testing samples are passed through the neural network several times, some weights are dropped each time, uncertainty is derived from variance across runs | Good |
| Tanno (17) | Differentiate among healthy brain, glioma, and multiple sclerosis | Diffusion tensor images or mean apparent propagator-MRI | Training: 16 subjects, validation: variable, overall 28 subjects | Convolutional neural network | Uncertainty-based classification correctly identified 96% of all high-risk (uncertain) predictions | External | Integrate intrinsic uncertainty with a heteroscedastic noise model and parameter uncertainty with Bayesian inference | Good |
| Valiuddin (18) | Density modeling of medical images | Thoracic computed tomography and endoscopic polyp images | 1,108 thoracic computed tomography scans, 1,000 polyp images | Probabilistic U-Net | Increased predictive performance (GED and IoU) of up to 14% with an approach that models uncertainty | External | Learn aleatoric uncertainty as a distribution of possible annotations using a probabilistic segmentation model | |
| Wang (33) | Classify diabetic macular edema | Optical cohere tomography images | 5,028 images | Convolutional and recurrent neural networks | Accuracy 0.951, F1-score 0.935–0.939, AUC 0.986–0.990 | External | Mean and standard deviation of probabilistic predictions yielded by ensemble of models | Good |
| Wickstrøm (47) | Classify polyps seen on colonoscopy | Images obtained from colonoscopies | 912 images | Fully convolutional network | IoU background = 0.946, IoU polyp = 0.587, mean IoU = 0.767, global accuracy = 0.949 | Internal | Monte Carlo dropout to approximate Bayesian posterior of weights, Monte Carlo-guided backpropagation, standard deviation of pixels | Good |
| Wieslander (19) | Investigate drug distribution on lung microscopy images | Rat lungs after treatment with different doses and routes of a medication | 1,105 images | Convolutional neural network | Precision = 0.89, recall = 0.87, F1 = 0.87; conformal prediction $R^2$ = 0.99 for actual vs. observed error | Internal | Conformal prediction using largest p-value minus second largest p-value | Good |
| *Non-imaging applications* | | | | | | | | |
| Cortes-Ciriano (24) | Drug discovery | Potency of a substance in inhibiting a biochemical or biological function | 24 protein drug targets, 203–5,207 bioactivity data points per protein | Ensembles of 100 deep neural networks | Strong correlation between confidence levels and percentage of confidence intervals encompassing true bioactivity ($R^2$ > 0.99, $p$<0.001) | External | Ensemble deep neural networks by recording network parameters throughout local minima during single network optimization, calculate variability and validation residuals across snapshots | Good |

(*Continued*)

**Table 1.** (Continued)

| Primary author | Purpose | Population or sampling unit | Sample size | Model architecture | Best model performance | Validation method | Method for quantifying prediction uncertainty | Quality Rating |
|---|---|---|---|---|---|---|---|---|
| Cortes-Ciriano (27) | Drug discovery | Potency of a substance in inhibiting a biochemical or biological function | 24 protein drug targets, 479–5,207 bioactivity data points per protein | Deep neural networks and random forest | Strong correlation between confidence levels and error rates ($R^2 > 0.99$, $p < 0.001$) | External | Conformal prediction to compute prediction errors on ensembles of predictions generated by dropout | Good |
| Scalia (25) | Predict molecular properties | Molecular graphs | 4 datasets: 130828, 103657, 11908, and 4200 graphs | Graph convolutional neural networks | Test set errors for 4 datasets: 0.74, 0.32, 1.33, 0.481 | External | Monte Carlo dropout, deep ensembles, and bootstrapping with comparison of these three methods | Good |
| Sieradzki (48) | Compound bioactivity prediction | Bit strings representing compound structures | Several sample sizes, largest: approximately 4,000 | Multi-layer perceptron | Models incorporating uncertainty information gained 0.004–0.007 precision | External | Pass test samples through the neural network serially, some weights dropped each time, uncertainty derived from variance in dropout | Good |
| Teng (28) | Predict Alzheimer's and Parkinson's disease progression | Clinical, imaging, genetic, and biochemical markers of neurodegenerative disease | Alzheimer's: 1,574 patients, Parkinson's: 1,093 patients | Deep generative model with recurrent neural networks | Alzheimer's: accuracy = 0.916, AUC = 0.981, F1 = 0.916; Parkinson's: accuracy = 0.797, AUC = 0.939, F1 = 0.797 | Internal | Ensemble of possible patient forecasts using a generative network | Good |
| Zhang (26) | Predict toxicity for chemical compounds | Toxicities of chemical compounds on nuclear receptors and stress response-related targets | Active class: 7039; inactive class: 89,922 | deep neural networks, random forest, light gradient boosting machine | Average AUC = 0.734; single-label predictions generated for about 90% of all instances with overall confidence 80% or greater | External | Conformal prediction using user-defined significance levels | Good |

AUC: area under the receiver operating characteristic curve, GED: generalized energy distance, IoU: intersection over union, MRI: magnetic resonance imaging.

variables. Once an epistemic uncertainty limit has been reached, quantifying the remaining aleatoric uncertainty in predictions could augment clinical application by allowing patients and providers to understand whether predictions have suitable accuracy and certainty for incorporation in shared decision-making, or are too severely compromised by aleatoric uncertainty to be useful, regardless of overall model accuracy [13]. These concepts are illustrated in **Fig 2**. This explanation considers transforming a given model into a stochastic ensemble through Bernoulli sampling of weights at model test time, giving rise to a measure of epistemic uncertainty for each sample.

## Medical imaging applications

Among the 24 studies describing medical imaging applications, 12 of those 24 (50%) used magnetic resonance imaging (MRI) features for model training and testing; 11 of those 12 (92%) of which involved the brain or central nervous system. The next most common sources of model features were retinal or fundus images (5 of 24, 21%) and endoscopic images of colorectal polyps (3 of 24, 13%). The remaining studies used computed tomography images, breast

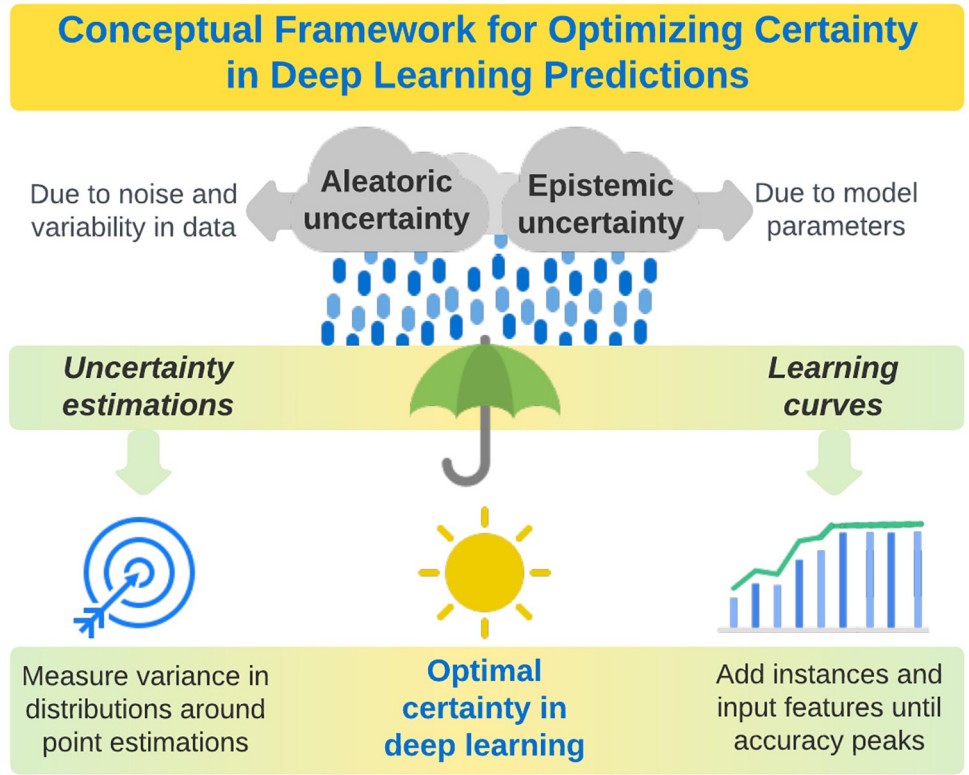

**Fig 2. A conceptual framework for optimizing certainty in deep learning predictions by quantifying and minimizing aleatoric and epistemic uncertainty.**

ultrasound images, lung microscopy images, or facial expressions. All model architectures included convolutional neural networks or a variation thereof (e.g., U-Net).

The predominant method for quantifying uncertainty in model predictions was Monte Carlo dropout, as originally described by Gal and Ghahramani as a Bayesian approximation of probabilistic Gaussian processes [14]. Briefly, during testing, multiple predictions are generated from a given network for which different neurons have dropped out. The neuron dropout rate is calibrated during model development according to training data sparsity and model complexity. Each forward pass uses a different set of neurons, so the outcome is an ensemble of different network architectures that can generate a posterior distribution for which high variance suggests high uncertainty and low variance suggests low uncertainty. Studies assessing the efficacy of uncertainty measurements provided reasonable evidence that uncertainty estimations were useful. In applying a Bayesian convolutional neural network to diagnose ischemic stroke using brain MRI images, Herzog et al [15] found that uncertainty measurements improved model accuracy by approximately 2%. In applying a convolutional neural network to estimate brain and cerebrospinal fluid intracellular volume, Qin et al [16] reported highly significant correlations (all $p<0.001$) between uncertainty estimations and observed error based on ground truth values. Finally, in applying a convolutional neural network for differentiating among glioma, multiple sclerosis, and healthy brain, Tanno et al [17] found that uncertainty-based classification correctly identified 96% of all predictions that had high-risk for error; this error was likely attributable to aleatoric uncertainty from noise and variability in data. Valiuddi et al [18] used Monte Carlo simulations in depicting the performance of a probabilistic U-Net performing density modeling of thoracic computed tomography and

endoscopic polyp images, learning aleatoric uncertainty as a distribution of possible annotations using a probabilistic segmentation model. This approach was effective in increasing predictive performance, measured by generalized energy distance and intersection over union, by up to 14%. Collectively, these findings suggest Monte Carlo dropout methods can accurately estimate uncertainty in predictions made by convolutional neural networks that make rare but potentially important misclassifications on medical imaging data, and corroborates prior evidence that Monte Carlo dropout can also offer predictive performance advantages, especially on external validation, by mitigating risk for overfitting.

Conformal prediction–used in two studies–demonstrated strong performance in estimating uncertainty. Wieslander et al [19] applied convolutional neural networks to investigate drug distribution on microscopy images of rat lungs following different doses and routes of medication administration, finding that conformal prediction explained 99% of the variance in predicted versus actual error. In another study by Athanasiadis et al [20], conformal prediction improved audio-visual emotion classification for a semi-supervised generative adversarial network compared with a similar network using the classifier alone.

Two studies used uncertainty estimation to compare modeling methods. Graham et al [21] used uncertainty measurements to demonstrate that a hierarchical approach to labeling regions and sub-regions of the brain produced similar predictive performance with greater certainty compared with a flat labeling approach, at any level of the labeling tree. Alternatively, to evaluate similarity between functional brain networks, Ktena et al [22] use convolutional neural network architectures in deriving a novel similarity metric on irregular graphs, demonstrating improver overall classification. Sedghi et al [23] calculated variance in displacement for different image classifications of brain MRIs, demonstrating good dice values for intra-subject pairs with consistent good results when simulating resections on the images, suggesting utility for challenging clinical scenarios.

## Non-imaging applications

The six studies describing non-imaging medical applications were heterogenous. Five of the studies endeavored to predict and classify biochemical and molecular properties for pharmacologic applications, each with somewhat different model architectures (i.e., ensembles of deep neural networks, convolutional neural networks, and multi-layer perceptrons). Three of these five studies generated posterior distributions and assessed variance across those distributions to approximate prediction uncertainty. In one instance, there was almost no gain in predictive performance; in another by Cortes-Ciriano and Bender, there was strong correlation between estimated confidence levels and the percentage of confidence intervals that encompassed the ground truth ($R^2 > 0.99$, $p<0.001$) [24]. This difference in performance may have been attributable to differences in model features. The less successful model used bit strings to represent molecular structures; the more successful model used high-granularity bioactivity features, with 203–5,207 data points per protein. A third study in the molecular property class also used Monte Carlo dropout techniques and reported relatively low test error values [25]. Two studies used conformal predictions to estimate uncertainty, one of which used conformal predictions in predicting active and inactive compound classes, generating single-label predictions for about 90% of all instances with overall confidence 80% or greater. Best results were demonstrated for deep neural networks rather than random forest or light gradient boosting machine models, and conformal prediction offered a controllable error rate and better recall for all three model types [26]. Cortes-Ciriano and Bender [27] leveraged conformal predictions in analyzing errors on ensembles of predictions generated by dropout, reporting strong correlation between confidence levels and error rates ($R^2 > 0.99$, $p<0.001$), with results similar to

those reported in their Deep Confidence work [24]. The remaining non-imaging study predicted neurodegenerative disease progression using multi-source clinical, imaging, genetic, and biochemical data, reporting variable predictive performance across different outcomes, but overall strong performance [28]. Compared with the biochemical prediction models, this study used a unique method for quantifying uncertainty, by measuring variance across predictions made by an ensemble of possible patient forecasts using a generative network. Collectively, these findings suggest that unique model architectures and methods for estimating uncertainty can be applied to a variety of non-pixel-based input features, producing occasional predictive performance advantages and accurate uncertainty estimations.

## Discussion

This review found that the uncertainty inherent in deep learning predictions are most commonly estimated for medical imaging applications using Monte Carlo dropout methods on convolutional neural networks. In addition, unique model architectures and uncertainty estimation methods can apply to non-pixel features, simultaneously improving predictive performance (presumably by mitigating risk for overfitting, in the case of Monte Carlo Dropout) while accurately estimating uncertainty. Unsurprisingly, for medical imaging applications, larger datasets of training images were associated with greater predictive performance [15,21,29–38]. We could not perform meta-analyses on predictive performance or uncertainty estimations because performance metrics and methods for quantifying uncertainty were heterogenous, despite relative homogeneity in model architectures–which were primarily based on convolutional neural networks–and homogeneity in methods for estimating uncertainty– which were primarily based on Monte Carlo dropout [14]. Uncertainty estimations for non-medical imaging applications were both sparse and heterogenous. Yet the weight of evidence suggests that a variety of methods can estimate uncertainty in predictions on non-pixel features, offering greater performance and reasonably accurate uncertainty estimations. Conformal prediction demonstrated efficacy in uncertainty estimation as well and is easy to interpret (e.g., at a confidence level of 80%, at least 80% of the predicted confidence intervals contain the true value), and applies not only to deep learning but also to other machine learning approaches such as random forest modeling.

For both imaging and non-imaging applications, uncertainty estimations are poised to augment clinical application by identifying rare but potentially important misclassifications made by deep learning models. First, mistrust of machine learning predictions must be overcome. Model explainability, interpretability, and consistency with logic, scientific evidence, and domain knowledge are critically important in building trust [7,8]. Yet, even when a model is easy to understand, generates predictions consistent with medical knowledge, and has 90% overall accuracy, patients and providers may wonder: is this prediction among the 1 in 10 that is incorrect? Can the model tell me whether it is certain or uncertain of this particular prediction? To address these questions and build trust, it seems prudent to include model uncertainty estimations in shared decision-making processes. Therefore, we believe that uncertainty estimations are a critical element in the safe, effective clinical implementation of deep learning in healthcare. In performing this review, we sought to summarize evidence regarding the efficacy of uncertainty estimation in building trust in deep learning among patients, caregivers, and clinicians, but we found little evidence thereof. Therefore, we propose uncertainty-aware deep learning as a novel approach to building trust.

We found no previous systematic or scoping reviews on the same topic, though several authors have described important components of estimating uncertainty in deep learning predictions. Common statistical measures of spread (e.g., standard deviation and interquartile

range) are undefined for single point predictions. Entropy, however, does apply to probability distributions. Therefore, most uncertainty estimation methods generate probability distributions around point estimations. Monte Carlo dropout, as originally described by Gal and Ghahramani, offers an elegant solution [14]. During testing, multiple stochastic predictions are generated from a given network for which different neurons have dropped out with specified probability. This dropout rate is calibrated during model development according to training data sparsity and model complexity. When training, dropping out different sets of neurons at different steps harbors the additional advantage of mitigating overfitting. When testing, each forward pass uses a different set of neurons; therefore, the outcome is an ensemble of different network architectures that can be represented as a posterior distribution. Variance across the distribution of predictions can be analyzed by several methods (e.g., entropy, variation ratios, standard deviation, mutual information). High variance suggests high uncertainty; low variance suggests low uncertainty.

This review was limited by heterogeneity in model performance metrics and methods for quantifying uncertainty. To identify the optimal methods for estimating uncertainty in deep learning predictions, it would be necessary to perform a meta-analysis or comparative effectiveness analyses. This would be facilitated by achieving consensus regarding core performance and uncertainty metrics. The field of deep learning uncertainty estimation is maturing rapidly; it would be advantageous to establish reporting guidelines, as has been done for prediction modeling, causal inference, and machine learning trials [39–42]. Finally, beyond uncertainty estimations, it may be useful to quantify how similar an individual patient is to other patients in the training data, so that users can understand whether uncertainty is attributable to variability in outcomes relative to similar features in the training data or due to a patient having outlier features that are not well represented in the training data.

## Conclusions

For convolutional neural network predictions on medical images, Monte Carlo dropout methods accurately estimate uncertainty. For non-medical imaging applications, a paucity of evidence suggests that several uncertainty estimation methods can improve predictive performance and accurately estimate uncertainty. Using uncertainty estimations to gain the trust of patients and clinicians is a novel concept that warrants empirical investigation. The rapid maturation of deep learning uncertainty estimations in medical literature could be facilitated by achieving consensus regarding performance and uncertainty metrics and standardizing reporting guidelines. Once standardized and validated, uncertainty estimates have the potential to identify rare but important misclassifications made by deep learning models in clinical settings, augmenting shared decision-making processes toward improved healthcare delivery.

## Supporting information

**S1 PRISMA Checklist. Preferred Reporting Items for Systematic Reviews and Meta-Analyses extension for Scoping Reviews (PRISMA-ScR) checklist.**
(DOCX)

## Acknowledgments

The content is solely the responsibility of the authors and does not necessarily represent the official views of the National Institutes of Health.

## Author Contributions

**Conceptualization:** Tyler J. Loftus, Benjamin Shickel.

**Data curation:** Tyler J. Loftus, Benjamin Shickel, Matthew M. Ruppert, Jeremy A. Balch, Tezcan Ozrazgat-Baslanti.

**Formal analysis:** Tyler J. Loftus, Benjamin Shickel.

**Funding acquisition:** Tyler J. Loftus, Benjamin Shickel, Parisa Rashidi, Azra Bihorac.

**Investigation:** Tyler J. Loftus, Benjamin Shickel, Matthew M. Ruppert, Jeremy A. Balch, Tezcan Ozrazgat-Baslanti, Patrick J. Tighe, Philip A. Efron, William R. Hogan, Parisa Rashidi, Gilbert R. Upchurch, Jr., Azra Bihorac.

**Methodology:** Tyler J. Loftus, Benjamin Shickel.

**Project administration:** Tyler J. Loftus, Philip A. Efron, William R. Hogan, Parisa Rashidi, Gilbert R. Upchurch, Jr., Azra Bihorac.

**Resources:** Tyler J. Loftus, Benjamin Shickel, Matthew M. Ruppert, Jeremy A. Balch, Tezcan Ozrazgat-Baslanti, Philip A. Efron, William R. Hogan, Parisa Rashidi, Gilbert R. Upchurch, Jr., Azra Bihorac.

**Software:** Tyler J. Loftus, Benjamin Shickel, Matthew M. Ruppert, Jeremy A. Balch, Tezcan Ozrazgat-Baslanti, Philip A. Efron, William R. Hogan, Parisa Rashidi, Gilbert R. Upchurch, Jr., Azra Bihorac.

**Supervision:** Patrick J. Tighe, Philip A. Efron, William R. Hogan, Parisa Rashidi, Gilbert R. Upchurch, Jr., Azra Bihorac.

**Visualization:** Tyler J. Loftus, Benjamin Shickel, Matthew M. Ruppert, Jeremy A. Balch, Tezcan Ozrazgat-Baslanti, Patrick J. Tighe, Philip A. Efron, William R. Hogan, Parisa Rashidi, Gilbert R. Upchurch, Jr., Azra Bihorac.

**Writing – original draft:** Tyler J. Loftus, Benjamin Shickel.

**Writing – review & editing:** Matthew M. Ruppert, Jeremy A. Balch, Tezcan Ozrazgat-Baslanti, Patrick J. Tighe, Philip A. Efron, William R. Hogan, Parisa Rashidi, Gilbert R. Upchurch, Jr., Azra Bihorac.

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
