## [Decision Letter · Decision Letter 0]

27 May 2022

PDIG-D-22-00112

­­­­­­­Uncertainty-aware Deep Learning in Healthcare: a Scoping Review

PLOS Digital Health

Dear Dr. Loftus,

Thank you for submitting your manuscript to PLOS Digital Health. After careful consideration, we feel that it has merit but does not fully meet PLOS Digital Health's publication criteria as it currently stands. Therefore, we invite you to submit a revised version of the manuscript that addresses the points raised during the review process.

One major question raised by a reviewer is a lack of novelty. Thus, we suggest the revised version could elaborate on this scoping review's novelty and research contribution. 

Please submit your revised manuscript by . If you will need more time than this to complete your revisions, please reply to this message or contact the journal office at digitalhealth@plos.org. Please include the following items when submitting your revised manuscript:

We look forward to receiving your revised manuscript.

Kind regards,

Yuan Lai, Ph.D.

Academic Editor

PLOS Digital Health

Journal Requirements:

1. Please provide a complete Data Availability Statement in the submission form, ensuring you include all necessary access information or a reason for why you are unable to make your data freely accessible. If your research concerns only data provided within your submission, please write "All data are in the manuscript and/or supporting information files" as your Data Availability Statement.

2. Please provide separate figure files in .tif or .eps format only and ensure that all files are under our size limit of 10MB.

For more information about how to convert your figure files please see our guidelines: https://journals.plos.org/digitalhealth/s/figures

Additional Editor Comments (if provided):

Reviewers' comments:

Reviewer's Responses to Questions

**Comments to the Author**

1. Does this manuscript meet PLOS Digital Health’s publication criteria? Is the manuscript technically sound, and do the data support the conclusions? The manuscript must describe methodologically and ethically rigorous research with conclusions that are appropriately drawn based on the data presented.

Reviewer #1: Yes

Reviewer #2: Partly

Reviewer #3: Yes

2. Has the statistical analysis been performed appropriately and rigorously?

Reviewer #1: N/A

Reviewer #2: N/A

Reviewer #3: N/A

3. Have the authors made all data underlying the findings in their manuscript fully available (please refer to the Data Availability Statement at the start of the manuscript PDF file)?

Reviewer #1: Yes

Reviewer #2: Yes

Reviewer #3: Yes

4. Is the manuscript presented in an intelligible fashion and written in standard English?

Reviewer #1: Yes

Reviewer #2: Yes

Reviewer #3: Yes

5. Review Comments to the Author

Reviewer #1: The authors address a highly relevant topic by conducting a scoping review on estimation and use of uncertainty around predictions by deep learning algorithms. In the abstract and introduction, they identify mistrust as a barrier to implementing deep learning in the healthcare setting, which I completely agree with. 

Major comments:

1) They set the goal to propose a conceptual framework to overcome this barrier, however this section in the Results rather reads like a summary of definitions and methods. I find the introduction of these highly relevant, but I think the concepts should be explained even more thoroughly (with examples), because I imagine that many in the target audience are not aware of these methods. With regard to this, I don't find Figure 1 very intuitive.

2) The explanation of epistemic and aleatoric uncertainty is somewhat confusing. In the abstract, the authors write "Overall, while uncertainty estimation accurately quantified aleatoric uncertainty", based on the Results section it seems like "Aleatoric uncertainty is difficult to address directly". Also, they authors claim that epistemic uncertainty can be reduced "by adding more training data", then they write that aleatoric uncertainty can be minimized by "broader data collection". It is not necessarily wrong logic, could be true for both, it just reads a bit confusing in contrasting the two uncertainties and highlighting the same solution.

3) I completely agree with the authors stating that comparing results is difficult due to different performance metrics across studies, and therefore I would tone down this part in the Results section and suggest to move sentences like "tended to have smaller sample sizes" etc. to the Discussion.

4) The methods are introduced to tackle mistrust, improvement of predictive performance is really highlighted in the Introduction, however in the Results, this aspect gets quite a bit of attention. Is there a difference between strong predictive performance of a model and strong performance in estimating uncertainty? It would be relevant to explain it to readers. Is improved performance e.g. with the MC method due to an ensemble effect? Some readers new to deep learning might understand this better with an example, analogy (decision tree - random forest, if this fits the purpose). 

5) In the title, the authors define the healthcare domain, therefore I suggest the exclusion of articles on oil prices, fruit images etc. Steinbrener (25), Akbari (39), Wang (41). They could be mentioned in the Discussion, but not as included articles in the scoping review.

Minor comments:

6) I suggest the use of the PRISMA-ScR Flow Chart in its original format and the inclusion of this in the main body of the article.

7) Reporting of frequencies and percentages could be slightly improved if written out more precisely, e.g. 11 out of 12 (92%) in the first sentence in Medical Imaging section that refers to another number (12) in that sentence as the 100% and not all 25 articles.

Reviewer #2: This study aimed to literature review of uncertainty-aware deep learning techniques for healthcare application. I have the following suggestions.

1. What is the novelty of this study although several literature reviews of Uncertainty-aware deep learning techniques for healthcare data have been done earlier?

2. The abstract should be rewritten and improved by combining the objectives, short methodology, main review findings, and prospective application.

3. Introduction section need to be improved. Recent studies related to State-of-art Uncertainty-aware ML/DL applications utilized in healthcare, need to extensively discussed.

4. Authors should add the several conceptual diagrams or figures to demonstrate the big picture of the scope of deep learning techniques in healthcare domain.

5. Authors should add a comparative summary the performance measures of the reviewed literatures.

6. Discussion section need to be extended and improved. Authors should discuss the strength and contradictories of reviewed findings in the discussion section.

Reviewer #3: This article addresses the issue of uncertainty quantification for deep learning predictive models.

These models are known to be very good in terms of predictive performance but also to lack explicability.

Therefore, in a delicate and critical domain such as healthcare, it is more than desirable to quantify the uncertainty associated to predictions even if the model error rate is very low.

For these reasons this manuscript is very interesting and it shows clearly various works in healthcare that are addressing the issue of uncertainty aware deep learning models.

The article is easy to read and to understand.

The subject of the article is in the scope of the journal,

There is no statistical analysis in the article, the authors explain clearly how they include articles in the review, and the table exposed on the articles give responses to the principle questions we can ask about the articles included on the scoping review.

Best regards,

Rayane ELIMAM

6. PLOS authors have the option to publish the peer review history of their article (what does this mean?). If published, this will include your full peer review and any attached files.

**Do you want your identity to be public for this peer review?** For information about this choice, including consent withdrawal, please see our Privacy Policy.

Reviewer #1: Yes: Adam Hulman

Reviewer #2: Yes: Iqram Hussain, PhD

Reviewer #3: Yes: Rayane ELIMAM

---

## [Decision Letter · Decision Letter 1]

9 Jul 2022

­­­­­­­Uncertainty-aware Deep Learning in Healthcare: a Scoping Review

PDIG-D-22-00112R1

Dear Dr. Loftus,

We are pleased to inform you that your manuscript '­­­­­­­Uncertainty-aware Deep Learning in Healthcare: a Scoping Review' has been provisionally accepted for publication in PLOS Digital Health.

Best regards,

Yuan Lai, Ph.D.

Academic Editor

PLOS Digital Health

Reviewer Comments (if any, and for reference):

Reviewer's Responses to Questions

**Comments to the Author**

1. If the authors have adequately addressed your comments raised in a previous round of review and you feel that this manuscript is now acceptable for publication, you may indicate that here to bypass the “Comments to the Author” section, enter your conflict of interest statement in the “Confidential to Editor” section, and submit your "Accept" recommendation.

Reviewer #1: All comments have been addressed

Reviewer #2: (No Response)

Reviewer #3: All comments have been addressed

2. Does this manuscript meet PLOS Digital Health’s publication criteria? Is the manuscript technically sound, and do the data support the conclusions? The manuscript must describe methodologically and ethically rigorous research with conclusions that are appropriately drawn based on the data presented.

Reviewer #1: Yes

Reviewer #2: (No Response)

Reviewer #3: Yes

3. Has the statistical analysis been performed appropriately and rigorously?

Reviewer #1: N/A

Reviewer #2: (No Response)

Reviewer #3: N/A

4. Have the authors made all data underlying the findings in their manuscript fully available (please refer to the Data Availability Statement at the start of the manuscript PDF file)?

Reviewer #1: Yes

Reviewer #2: (No Response)

Reviewer #3: Yes

5. Is the manuscript presented in an intelligible fashion and written in standard English?

Reviewer #1: Yes

Reviewer #2: (No Response)

Reviewer #3: Yes

6. Review Comments to the Author

Reviewer #1: I think the revised version has made the suggested framework more clear. I have no further comments.

Reviewer #2: This manuscript need to improved in a great extent.

Reviewer #3: This article is very interesting and clearly addresses the issue of uncertainty in deep learning for certain application.

As i point in the first review this question should be adressed and more explored because of the high number of application which use Deep learning in healthcare.

Therefore the prediction uncertainty quantification is necessary in this context

Minor comment:

Perhaps the notions of aleatoric and epistemic uncertainty should be explained more clearly.

7. PLOS authors have the option to publish the peer review history of their article (what does this mean?). If published, this will include your full peer review and any attached files.

**Do you want your identity to be public for this peer review?** For information about this choice, including consent withdrawal, please see our Privacy Policy.

Reviewer #1: **Yes: **Adam Hulman

Reviewer #2: No

Reviewer #3: No
